# Lowering the Intraocular Pressure in Rats and Rabbits by *Cordyceps cicadae* Extract and Its Active Compounds

**DOI:** 10.3390/molecules27030707

**Published:** 2022-01-21

**Authors:** Li-Ya Lee, Jui-Hsia Hsu, Hsin-I Fu, Chin-Chu Chen, Kwong-Chung Tung

**Affiliations:** 1Department of Veterinary Medicine, National Chung Hsing University, Taichung 402204, Taiwan; ly.lee@grapeking.com.tw; 2Biotech Research Institute, Grape King Bio Ltd., Taoyuan 325002, Taiwan; juihsia.hsu@grapeking.com.tw (J.-H.H.); hsini.fu@grapeking.com.tw (H.-I.F.); 3Institute of Food Science and Technology, National Taiwan University, Taipei 104336, Taiwan; 4Department of Food Science, Nutrition and Nutraceutical Biotechnology, Shih Chien University, Taipei 104336, Taiwan; 5Department of Bioscience Technology, Chung Yuan Christian University, Taoyuan 320314, Taiwan

**Keywords:** *Cordyceps cicadae*, intraocular pressure (IOP), ganglion cells, N6-(2-hydroxyethyl)-adenosine (HEA)

## Abstract

*Cordyceps cicadae* (CC), an entomogenous fungus that has been reported to have therapeutic glaucoma, is a major cause of blindness worldwide and is characterized by progressive retinal ganglion cell (RGC) death, mostly due to elevated intraocular pressure (IOP). Here, an ethanolic extract of *C. cicadae* mycelium (CCME), a traditional medicinal mushroom, was studied for its potential in lowering IOP in rat and rabbit models. Data showed that CCME could significantly (60.5%) reduce the IOP induced by microbead occlusion after 56 days of oral administration. The apoptosis of retinal ganglion cells (RGCs) in rats decreased by 77.2%. CCME was also shown to lower the IOP of normal and dextrose-infusion-induced rabbits within 60 min after oral feeding. There were dose effects, and the effect was repeatable. The active ingredient, N6-(2-hydroxyethyl)-adenosine (HEA), was also shown to alleviate 29.6% IOP at 0.2 mg/kg body weight in this rabbit model. CCME was confirmed with only minor inhibition in the phosphorylated myosin light chain 2 (pMLC2) pathway.

## 1. Introduction

*C. cicada* is an entomopathogenic fungus that produces fruit on the head of the host (*Cicada flammata*) and then forms a fruiting body on the surface of the insect. The hybrid of fruiting body and insect has been used as traditional Chinese medicine for 1600 years. It possesses many medical effects similar to *Hirsutella sinensis* such as antioxidation [1,2], anti-inflammation [3,4], lowering blood sugar [5], and renal protection [6,7]. Zheng’s findings indicated that HEA, an active compound from *C. cicadae*, has a beneficial effect on UUO-induced tubulointerstitial fibrosis by suppressing inflammation and renal fibroblast activation. Inhibiting renal fibrosis of *C. cicadae* in vivo may be through the TGF-β1/CTGF pathway [8]. Treatment with HEA (20 and 40 mg/kg) for six weeks reduced blood glucose in alloxan-induced diabetic rats [9]. The safety of *C. cicadae* mycelium has been evaluated in rat, rabbit, pig, and mouse models, as well as humans [10,11,12,13,14,15,16,17,18,19]. 

Recent studies on novel applications of *C. cicadae* have focused on eye health. *C. cicadae*-fermented mycelia extract in a benzalkonium chloride (BAC)-induced dry eye model has been evaluated in mice. *C. cicadae* extract effectively ameliorated BAC-induced dry eye symptoms via enhancement of cornea resilience against BAC (10 mg/kg)-induced damage and maintenance of conjunctival goblet cells [20]. In a steroid-induced high IOP rat model of glaucoma, elevated IOP levels decreased significantly after four weeks of oral treatment with water or ethanol extracts of *C. cicadae*. The results were almost as efficient as the Alphagan positive control. There were significant decreases in malondialdehyde and lactate dehydrogenase levels after 28 days of administration parallel to IOP decreases and significant increases in catalase, superoxide dismutase, and glutathione peroxidase levels [21]. Glaucoma is a neurodegenerative disease characterized by the loss of retinal ganglion cells (RGCs) [22]. Inhibition of immune activity in the retina (microglia) may decrease RGC death. This study evaluated IOP decreases in animal models and then identifies the active ingredient.

## 2. Materials and Methods

### 2.1. Ethanol Extract of C. cicadae Mycelium (CCME) Preparation 

*C. cicadae* (BCRC MU 30106) deposited in Bioresources Collection and Research Center (BCRC) in Food Industry Research and Development Institute (Hsinchu, Taiwan) was incubated in a medium composed of 2% glucose, 1% yeast extract, and 1% soybean powder adjusted to pH 6.0 at 25 °C on a rotary shaker (120 rpm) for 3 days. The fermentation process was scaled up from 1.0 L into 200 L fermenter with the same medium for 5 days and then was transferred into a 20-Ton fermenter (working volume 16 Ton) for 7 days in its final production. After culturing for seven days at 25 °C, the broth was heated to 100 °C for 1 h and concentrated in vacuo at 55 °C; it was then freeze dried, ground to a powder, and stored at 4 °C for use. The C. cicadae mycelium powder was extracted with 95% ethanol (1:10, *w*/*v*) at room temperature for 1 h. The extract was filtered through Whatman No.4 filter paper, and the rotor was evaporated to obtain the dried extract (CCME). CCME was suspended into various concentrations in soybean oil/water (1:1) mixture and gavaged into rabbit and rat at 1.25 mL/kg body weight.

### 2.2. High-Performance Liquid Chromatography (HPLC) Analysis

Analysis was performed on HPLC (Hitachi 5000 series system) equipped with UV wavelength detector set at 254 nm, and a reverse-phase column (Inert^®^, ODS-2, 250 mm × 4.6 mm, 5 μm) was used with a column oven at 40 °C. The mobile phase was composed of water (A) and acetonitrile (B) with a gradient as follows: 0–15 min 100% A; 15–40 min 100–80% A; 40–55 min 80–5% A; 55–75 min 50–0% A; 75–90 min 0% A; 90–91 min 0–100% A; 91–100 min 100% A. The flow rate was kept at 1.0 mL/min, and the injection volume was 10 μL. Identification of adenosine and HEA was determined by comparing retention times with the standards (Sigma-Aldrich, St. Louis, MO, USA). CCME was dissolved in 20-fold volumes of 95% ethanol and filtered through a 0.45 μm filter before HPLC analysis.

### 2.3. Liquid Chromatography–Quadrupole-Time of Flight Mass Spectrometer (LC–QTOF/MS) Analysis

The mass spectrometer was operated in the positive ion mode (QQQ/MS model: API 3000, Applied Biosystem, Waltham, MA, USA). The nebulizer gas, curtain gas, collision gas, ion spray voltage, and source temperature were set at 8 psi, 7 psi, 2 psi, 4500 volts, and 350 °C, respectively. The mobile phase was composed of water (A) and acetonitrile (B). Separation was optimized using a gradient method with mobile phase A/B set to 95%/5% from 0.00 to 5.00 min and 0%/100% from 5.00 to 10.00 min and then back to 95%/5% from 13.50 to 15.00 min. the flow rate was 300 μL/min, and the sample injection volume was 10 μL. The mass spectrometer was operated in the positive ion mode. The nebulizer gas, curtain gas, collision gas, ion spray voltage, and source temperature were 8 psi, 7 psi, 2 psi, 4500 volts, and 350 °C, respectively.

### 2.4. Microbead Occlusion in Rat Model

Male SD rats were obtained from the National Laboratory Animal Center (Taipei, Taiwan). The rats (170–200 g) were maintained in rat cages with standard atmospheric conditions of 12 h light and dark periods at 25 ± 0.5 °C with a relative humidity of 60 ± 5%. All methods were carried out in accordance with relevant guidelines and regulations and were approved by the ethical committee of Hualien Tzu Chi Hospital (106-02). A 10 μL suspension containing 30 mg/mL gamma-irradiated 8 μm magnetic microspheres in Hank’s balanced salt solution (HBSS) was injected into the anterior chamber of the left eye via a 33-gauge bevel needle. The IOP was measured five times with the Icare^®^ TONOLAB tonometer (iCare, Vantaa, Finland) before injection and daily after use. CCME (100 mg/kg/b.w.) was gavaged once daily for 56 days. All data are expressed as mean ± SEM. All frozen sections of the retinas were cut into 1 to 2 mm thickness from the ON head, to ensure the use of equivalent fields for comparison [23]. A TUNEL assay (DeadEndTM Fluorometric TUNEL System, Promega Corporation, Madison, WI, USA) was performed to detect apoptotic cells. The TUNEL-positive cells in the RGC layer of each sample were counted in 10 high-powered fields (HPF, 400× magnification).

The detailed procedure has been previously described in a previous report (Tsai et al. 2008). At least five randomly chosen areas (each 62,500 μm^2^) in the central and mid-peripheral regions of each retina and their averages were estimated as the mean density of RGCs per retina. The retinas were examined with a 400× epi-fluorescence microscope (Axioskop; Carl Zeiss Meditech, Inc., Thornwood, NY, USA) equipped with a filter set (excitation filter = 350–400 nm; barrier filter = 515 nm), a digital camera (Axiocam MRm), and software (Axiovision 4.0). All data are expressed as mean ± SEM.

### 2.5. ROCK Kinase Assay

The ROCK kinase assay is a luminescent kinase assay that measures IOP formed from a kinase reaction. ADP is converted into ATP and is converted into light by Kinase-Glo Luciferase. The Kinase-Glo Assay can be used to monitor the activity of virtually any ADP-generating enzyme. The relative light unit (RLU) without test compound was set as 100% (blank value), and that without enzyme and compound was set as 0% (normal value). The reaction rate (% of black) was then calculated from the RLU. 

The following reagents were used in this assay: ROCK1 (750 ug/mL, Carna Biosciences, Cat.01-109); long S6 kinase substrate peptide (Millipore, Cat#12-420); magnesium chloride solution (Sigma, 60142-100 mL); EGTA solution (0.5 M pH 8.0, Bioword, Cat. 4052008-1); Trizma^®^ hydrochloride solution (1 M, pH 7.5 Sigma, T2319-100 mL); bovine serum albumin solution (20 mg/mL in water, Sigma); Kinase-Glo luminescent kinase assay (Promega, RV6712); ripasudil (K115) hydrochloride dehydrate (Selleckchem, S7995-5 mg); ATP solution (100 mM, GE Healthcare, Cat.27-2056-010). A ROCK kinase inhibitor k-115 was used as the positive control.
(1)$$\mathrm{Inhibition}\;\left(\%\right)=\frac{100\;–\{\left(\mathrm{activity}\;\mathrm{of}\;\mathrm{Enzyme}\;\mathrm{with}\;\mathrm{Test}\;\mathrm{Cmpd}\;\&\;\mathrm{Sbustrate}-\mathrm{Min}\right)\}\times 100}{\mathrm{Max}-\mathrm{Min}}$$

Max = observed enzyme activity measured in the presence of enzyme, substrate(s), and cofactors utilized in the method.

Min = Normal value in the method

### 2.6. MLC Phosphorylation Inhibition (p MLC) Assay

Myosin light chain2 (MLC2) is also known as myosin regular light chain (MRLC). ROCK could phosphorylate ser19 of MLC2 of smooth muscle. ROCK inhibitors could lower IOP by increasing aqueous humor outflow through the trabecular meshwork (TM). The A7r5 cell (Rat muscle cell, ATCC CRL-1444) was purchased from American Type Culture Collection (Manassas, VA, USA) and was maintained in DMEM medium supplemented with 10% fetal bovine serum and 1% penicillin–streptomycin antibiotics at 37 °C in a humidified atmosphere of 5% CO_2_. Cells were seeded at 6*10^5^ cells per well in a P100 dish and incubated overnight.

Cells were changed to a new medium prior to treatment. Cell lysis and Western blots were performed by following the usual protocol. CCME prepared in 100% DMSO as a stock solution was added to cell lysate at various concentrations. AR-13324 was used as the positive control. The cell lysate was assayed with pMLC2 (Thr19/Ser19) antibody (Cell Signaling, CST3674) in HaltTM protease and phosphatase inhibitor (Thermo Fisher 78442, Waltham, MA, USA). GAPDH (Sigma), HRP goat anti-rabbit (Jackson Immuno Research, 111-035-003, West Grove, PA, USA), skim milk, and SuperSignal West Pico Maximum Sensitivity substrate (Thermo Fisher, 34080) were used for the Western blot.

### 2.7. Normal Rabbit Model

For this study, 8–12-week-old female New Zealand White (NZW) rabbits were purchased from HUEI-JYUN Company (Changhua, Taiwan). The animals were housed in the rabbit laboratory of Master Laboratory Co., Ltd. During the study, data of the animals including cage number, strain, weeks old, animal I.D., date of animal arrival, experimental number, group, and period were noted on the housing card. The housing conditions were kept in 12 h light and 12 h dark at 23 ± 2 °C and 40–70% humidity. The animals were given free access to food and water. To ensure the health of the animals, clinical observations were noted daily by veterinarians from Master Laboratory Co., Ltd. and Industrial Technology Research Institute (ITRI), Hsinchu, Taiwan, during quarantine and experiments, respectively. The animals were placed in this study after one week of quarantine. The experimental protocol listed below has been reviewed and approved by the Institutional Animal Care and Use Committee (IACUC-2016-040) of ITRI. In total, 25 rabbits were randomly divided into five groups (blank, 0.5% Timolol via eye drop; vehicle, 2.5 mg/kg/b.w. and 25 mg/kg/b.w. CCME oral); one drop of 0.5% Timolol (Timoptol^®^ Ophthalmic Solution 0.5%: 0.5% timolol maleate, MSD) was applied to the right eye of each rabbit per day as a positive control.

The IOP of the right eye was measured five times with Tono Vet Tonometer type Tv01.
ΔIOP = IOP time point − IOP 0 h(2)
Percentage of ΔIOP = ΔIOP/IOP 0 h(3)

Data are presented as the mean ± S.E.M. (standard error of mean). Differences between groups were analyzed with Student’s two-tailed *t*-test; *p* < 0.05 was considered significant.

### 2.8. Dextrose-Induced Acute Glaucoma Model

Three male (2.8–3.0 kg) and three female (3.0–3.6 kg) New Zealand white rabbits were purchased from Weishinhun company (New Taipei City, Taiwan) and were quarantined and housed under standard conditions (12 h dark, 12 h light, 20 °C, 55 ± 15% humidity) for seven days before the experiment. All producers were conducted under the SOP (SOPA-303, SOPA-203, SOPA-315, and SOPA-206) of SuperLab (New Taipei City, Taiwan). Rabbits were given free access to Altromin 2023 diet and R.O. water. Every rabbit had its own cage. They were randomly divided into three groups—one male and one female per group. The same dose of soybean oil/water mixture was given as the control group. IOP was measured with Tono-Pen VETTM Tonmeter type without anesthesia. Here, 5% glucose solution (15 mL/kg body weight) was injected through the marginal ear vein to rapidly induce ocular hypertension. A repeat experiment was conducted with the same rabbits after a one-week washout. Values are expressed as the means ± standard deviation (SD). The Mann–Whitney U test was used to compare the mean value between the control group and the CCME-treated group. Statistical significance was defined as a *p*-value < 0.05. 

## 3. Results

### 3.1. Glaucoma Model

Normal rat IOP values are 10–12 mmHg. A 10 uL magnetic microbeads suspension was injected into anterior chambers of the right eye to block the outflow of the aqueous humor (AH) at day 0 (Figure 1). The IOPs of both control and CCME groups before surgery were recorded as 10.8 ± 0.91 and 10.8 ± 1.22 mmHg, respectively. After surgery, 1.25 mL/kg b.w. CCME and vehicle were gavaged on the same day and for the subsequent 56 days once daily. On day 1, a substantial increase in IOP from day 0 in both control (25.4 ± 3.2 mmHg) and CCME groups (23 ± 1.52 mmHg) were observed. The CCME group IOP was lower than the control group, but the difference was not significant. Continuously higher IOPs in the control were seen from day 1 to day 56 (25.4 ± 3.2 mmHg) versus the CCME. Compared with the sustained high IOPs of control, the CCME had a lower IOP (23 mmHg) at day 2 (22.1 ± 1.91 mmHg). This further decreased the IOP on day 3 (18.2 ± 2.04 mmHg) versus the control group (*p* < 0.05). The IOP decreased and remained at 17.3 ± 1.70 mmHg for the next 53 days. These data suggest a significant alleviation of CCME when IOP increases. The ΔIOP between control and CCME groups progressively increased, but the alleviated IOP in CCME still did not reach baseline values (10.8 mmHg) even after 56 days. On day 56, the IOP of the CCME group was 60.5% lower than the control group (26.5–17.3 mmHg)/(26.5–10.8 mmHg) mmHg).

There was mass mortality due to the impairment of microbeads. Terminal deoxynucleotidyl transferase (TdT) dUTP nick-end labeling assay was performed using a microscope (Figure 2a). The corresponding IOP reduction led to apparent protection of CCME in RGCs mortality versus control. Nearly 77.2% of RGCs were protected from death (Figure 2b).

### 3.2. ROCK Kinase Assay

The IOP reduction may be caused by a decrease in AH outflow resistance or a decline in AH formation. Rho-associated kinase (ROCK) is a critical serine/threonine kinase Rho GTPase. ROCK inhibitions were reported to have a lower IOP value versus increasing the aqueous outflow through the TM. Rock kinase inhibition and myosin light chain (MLC) phosphorylation confirmed the inhibitor activity in CCME.

Both doses of CCME (0.25 and 2.5 μg/mL) showed dose-dependent but not significant inhibition (4 ± 4 and 29 ± 4%) versus K-115 (M.W. 323.4; Table 1). This positive control reduced ROCK kinase activity at IC50:21.4 ± 1.6 nM. Figure 3 compares the abilities of controls and CCME to reduce MLC phosphorylation level in the cell line model. A high dose (25 μg/mL) of CCME could reduce 59.0% phosphorylation and was dose-dependent but had poor potential versus IC50: 1 nM for AR-13324 (M.W.526.4).

### 3.3. Normal Rabbit Model

NZW rabbits with a normal IOP were tested to realize an IOP-reduced effect of CCME on cross-species animals. Timolol eye drops served as a positive control. CCME was fed orally, and IOP was measured at 0, 1, 3, and 24 h on day 1 and day 2 to match Timolol’s optimal function time. Both Timolol and high dose CCME immediately lowered the IOP at 1 h (−2.5 and −1.5 mmHg), with a slight increase at 3 h. This then returned to baseline IOP at 24 h (Figure 4a,b).

The repeatability of IOP decrease with an increase in back-to-normal conditions on day 2 was observed in both Timolol and CCME groups. CCME also had a dose dependence on day 1 and day 2. Milder IOP decreases (12%) were seen for high-dose CCME. This is less effective than 20% for Timolol at 1 h, but the convenience of the oral route and the quick responses still potentialized these applications.

### 3.4. Dextrose-Induced Acute Glaucoma Model 

We pathologically studied acute high IOP rabbit models induced by posterior intravenous administration of 5% dextrose infusion. The IOP sharply increased, from 23.7 to 38.4 mmHg (*n* = 4), at 15 min after dextrose infusion and then gradually decreased to initial IOP at 90 min (Figure 5). CCME requires one hour for activity (Figure 4); the pretreatment of CCME at −60 min was designed to function between 0 and 90 min. Compared with the vehicle, CCME obviously displayed less of a rise in IOP from 15 to 90 min. Here, a 25.96% reduction (37.75→29.95) in IOP was seen with a 53.35% reduction (29.95–23.4)/(37.75–23.4) in IOP elevation at 15 min for high-dose CCME. There was less area under the curve (AUC) versus control. For CCME, the IOP at time 0 min before infusion was nearly the same as that at time −60 min, indicating that the time required for CCME is at least 60 min.

### 3.5. CCME Active Components

The fingerprint chromatogram contains two specific active components purified from the liquid fermentation of CCME (retention time 27.7 min and 30.8 min), as shown in Figure 6a. These compounds’ structures were confirmed using the liquid chromatography in combination with tandem MS (LC-QTOF/MS) operating in multiple reaction monitoring (MRM) mode by identifying precursor→production transitions (Figure 6b). 

Concentrations of adenosine and HEA were determined as 2.2 and 5.1 mg/g, respectively. Adenosine has been shown to have low bioavailability due to fast metabolism in vivo [24]. HEA is an adenosine derivative and has anti-hydrolytic capability from adenosine deaminase (ADA) hydrolysis (data not shown). HEA was chosen as a potential candidate to assess its IOP reducing activity. HEA significantly mitigated the IOP increase similar to CCME (Figure 7). Relative to the vehicle, there was a nearly 26.04% reduction (38.4→28.4) in IOP and 68.5% reduction (28.4–23.8)/(38.4–23.8) in IOP elevation at 15 min in both eyes in the high dose group (0.2 mg/kg body weight). Dose-dependent effects were also noted at all sampling times.

## 4. Discussion

According to the 2020 WHO study, almost one billion people are afflicted with vision impairment including glaucoma, retinal inflammation, dry eye, diabetic retinopathy, and age-related macular degeneration [25,26]. Glaucoma is the second most frequent cause of irreversible blindness in the world and the global burden is predicted to be 112 million in 2040 [27]. Glaucoma is a progressive loss of RGCs, optical nerve head, and retinal nerve filter layers resulting in visual field deficiencies. Glaucoma caused by optical nerve damage is related to IOP increase. Axon degeneration of RGCs at the optic nerve head results from high IOP and is parallel to the apoptosis of RGCs. This is believed to be a leading risk factor for the onset and progress. Lowering IOP can protect the nerve and reduce the development and progression of glaucoma.

*C. cicadae* mycelium extracts (water and ethanol) can significantly reduce the steroid-induced high IOP in rats similar to Alphagram. This study confirmed the IOP-dropping effect in various models and animal species. The microbead occlusion model was used to induce ocular hypertension in rats. A microbead suspension was injected into the anterior chamber to occlude aqueous humor outflow and to raise the IOP 135.1% (from 10.8 to 25.4 mmHg) at day 1 for the control group. Compared with control, CCME decreased the IOP (25.4 ± 3.9 mmHg) at day 1; the IOP gap between both groups continued to increase until day 3 (26.4 ± 18.2 mmHg) and stabilized until day 56 (26.5 ± 1.95 mmHg vs. 17.3 ± 1.7 mmHg). 

During these 56 days, the IOPs in CCME-treated animals were always higher than those of initial IOP (10.8 ± 1.22 mmHg) on day 0 before bead injection. Microbeads are difficult to degrade, and these data implied that CCME’s partial dropping function may arise from creating more leakages between the bead and trabecular meshwork (TM) cells or reducing aqueous humor (AH) production by the ciliary body epithelium. Glaucoma is a chronic neuropathy mostly associated with elevated IOP. In this experiment, there were 56 days between model induction and RGC impairment under high IOP. After sacrifice at day 56, dead RGCs were evaluated with a TUNEL assay. A higher IOP indeed led to much more apoptosis in RGCs in the vehicle and CCMC groups than in the sham group (Figure 2). According to the report of Nakazawa et al. (2006), elevated IOP would stimulate the Fas ligand binding to the death receptors and result in apoptosis in RGCs [27]. The main cause of vision loss in glaucoma is apoptosis in RGCs. Significant reductions in CCME in apoptosis are attributed to lower IOP values. CCME protected 77.2% RGCs from mortality by mediating a lower IOP versus control. Bradely et al. (1998) discovered the contractile properties of TM cells. Increasing the contraction of TM cells in glaucoma may result in elevated AH outflow resistance [28]. Disruption of TM actin cytoskeleton resulting in reducing outflow resistance has also been observed by Tian et al. [29,30]

Rhopressa^®^ (netarsudil ophthalmic solution, Aerie Pharmaceuticals) is a rho kinase (ROCK) inhibitor and was approved by the FDA in late 2017 for reducing IOP in primary open-angle glaucoma or ocular hypertension patients by increasing the AH outflow. Its molecular mechanism was believed to decrease actin–myosin contraction, with reduced actin stress fibers and focal adhesion in TM [26]. 

CCME was evaluated with ROCK assay to confirm its possible inhibitory mechanism. Table 1 and Figure 3 show that no significant inhibition was found for CCME even up to 25 mg/mL. This inhibiting concentration is far more than those of positive controls. Moreover, CCME was given orally. The concentration would be further diluted before it reached TM cells in vivo. These data implied that the IOP decreasing effects are not caused by ROCK inhibition. 

Adenosine receptors (ARs) are well repressed in all the ocular tissues such as TM and retina. ARs are implicated in retinal function, neuronal survival, and blood flow. Many studies have focused on the role of ARs on AH formation and outflow facility, IOP, and optic nerve protection. AR agonists and antagonists have also been reported to modulate aqueous humor formation, outflow facility, IOP modulation, and optic neuroprotection [31]. Some of these are exploited in clinical trials for glaucoma and dry eye disease [32]. Many AR agonists and antagonists have been evaluated as potential therapeutic agents in glaucoma. However, there are different effects of the AR agonists and antagonists in different animal species. The cross-species difference should be evaluated. 

Aside from rats, normal NZW rabbits were fed with CCME to check their IOP mitigation. Timolol is a β-adrenergic antagonist (beta blocker). These eye drops can treat glaucoma, and therefore, they served as a positive control. Timolol achieved its maximal IOP-lowering effect at 1–2 h after administration, and thus, sampling time was set at 0, 1, 3, and 24 h. CCME had its optimal IOP reduction at 1 h with dose dependence and reproducibility for 2 days (Figure 4). It was quickly available orally. It was not as effective as Timolol, but the oral delivery has advantages in terms of compliance and side effects from preservatives in eye drops. In addition, the active ingredient purified from CCME might be used in targeted delivery. 

In addition to normal rabbits, the pathologic acute high IOP rabbits induced by dextrose infusion were also studied (Figure 5). In all three groups, dextrose promptly raised the IOPs within 15 min with a gradual decline to baseline in 90 min. CCME also showed significant prevention and dose dependence. At 1 h after administration, it is necessary for CCME to exhibit an IOP decrease in normal rabbits (Figure 5); pre-treating with CCME at −60 min was carried out before dextrose induction. The IOPs at −60 min were almost the same as those at 0 min before dextrose injection for all placebo and CCME groups. This suggests that 60 min is required for this activity. 

The CCME ethanol extract was rich in adenosine and HEA (adenosine derivative) (Figure 6). We have never found cordycepin in C. cicadae mycelium or fruiting bodies. The roles of adenosine in IOP modulation have been well established including binding to A1AR of TM cells to reduce outflow resistance in lowering IOP and stimulating A3AR to increase the fluid inflow of AH and IOP [33,34]. HEA was studied because it is a known Ca^2+^ antagonist in inhibiting muscle contraction. The HEA structure may be legitimately considered as an AR agonist or antagonist. The assessment was conducted with the same induced model as CCME. As a result, HEA showed a significant IOP reduction with dose dependence (Figure 7). Santiago et al. (2002) mentioned that ARs exist in the ciliary body, trabecular meshwork, scleroderma, and retina. They are related to AH formation, outflow, and IOP homeostasis [35]. Zhong et al. found that neuronal survival and blood flow were also affected by ARs. 

Many A1AR agonists, A2aAR agonists, and A3AR agonists are potentially therapeutic candidates for treating inflammatory diseases such as glaucoma by decreasing IOP and inducing specific anti-inflammatory effects [36,37]. We speculated that HEA may mediate significant effects on IOP similar to AR agonists or antagonists [38,39,40,41]. As an adenosine derivative, HEA may function as an AR agonist or antagonist in decreasing IOP and preventing RGC death in rats and rabbits. This hypothesis was partially supported by HEA activation of A1AR and A2aAR as agonists [42,43]. Activated P2X7 receptor on RGCs increases intracellular Ca^2+^ levels and induces cell apoptosis. Adenosine and ATP are agonists of the P2X7 receptor and can activate P2X7 receptors [44]. We do not know the effect of HEA on the P2X7 receptor, but it is known as a Ca^2+^ channel antagonist and can block the influx of Ca^2+^ into the cytosol. Recently, HEA was found in rat brain after 30 min of i.v. administration. This proved that HEA could pass through the blood–brain barrier (BBB). The CCME promptly initiated IOP reduction at 1 h following the oral route [19]. Studies on pharmacokinetics and blood–retina–barrier (BRB) are also needed.

## 5. Conclusions

*Cordyceps cicadae* mycelium alcohol extract (CCME) was shown to have IOP-reducing potential in rat and rabbit models. This study was confirmed with only minor inhibition in the phosphorylated myosin light chain 2 (pMLC2) pathway. Its active ingredient was HEA, which could significantly mitigate the IOP. CCME also protected RGCs from death, similar to the high IOP in rats. Future efforts will focus on elucidating the mechanism and possibly achieving the regulate intraocular pressure.

## Figures and Tables

**Figure 1 molecules-27-00707-f001:**
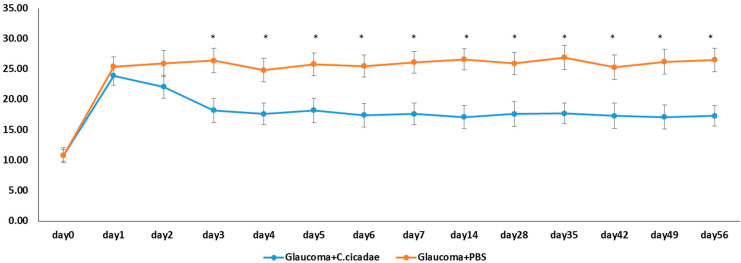
Effect of C. cicadae mycelia extract on IOP in the glaucoma model. *: *p* < 0.05, *n* = 12, mean ± SEM. versus Glaucoma+PBS group.

**Figure 2 molecules-27-00707-f002:**
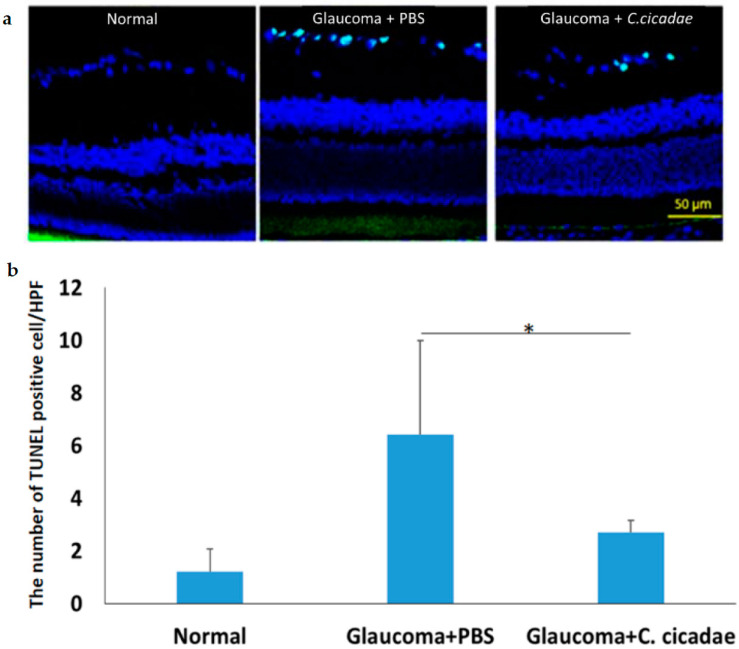
The apoptosis of retinal ganglion cells (RGCs) of rats at day 56. (**a**) Representative terminal deoxynucleotidyl transferase dUTP nick-end labeling (TUNEL) staining of day 56. (**b**) The number of TUNEL-positive cells in the RGC. (mean ± standard error of the mean, SEM; * *p* < 0.05).

**Figure 3 molecules-27-00707-f003:**
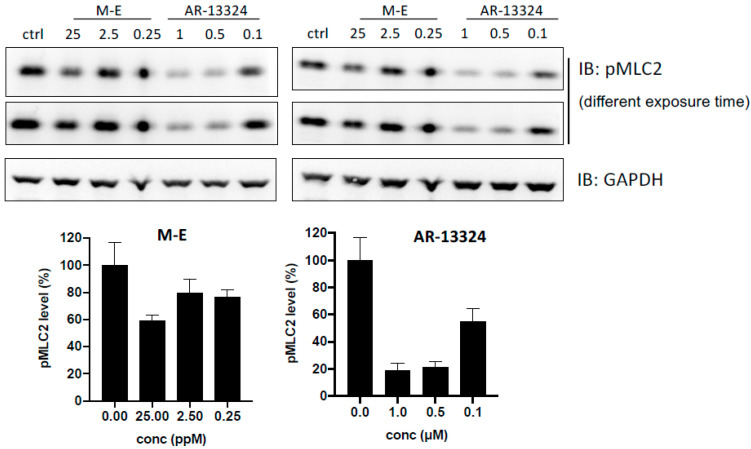
pMLC2 inhibition for CCME and ROCK inhibitor AR-13324 by Western blot.

**Figure 4 molecules-27-00707-f004:**
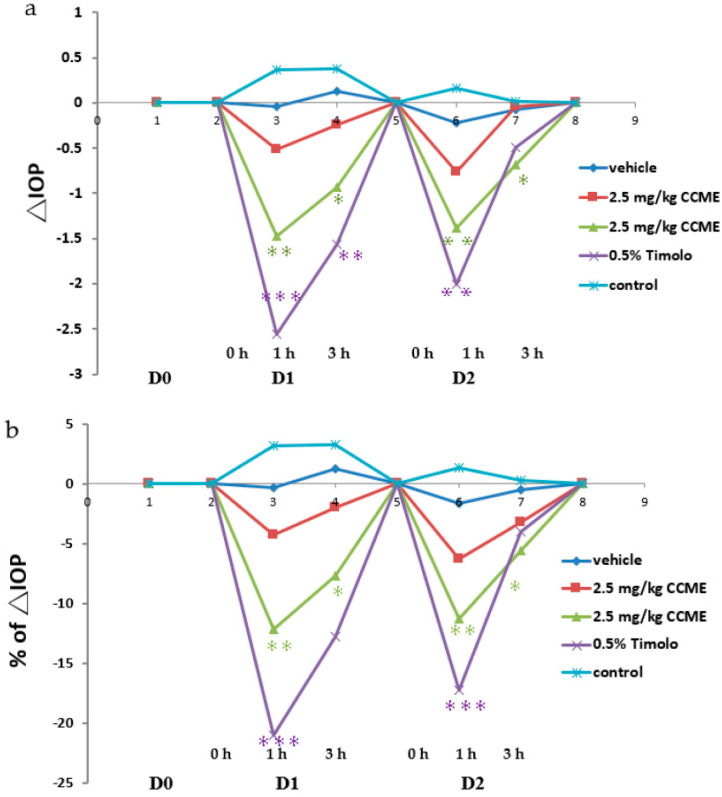
Effect of *C. cicadae* mycelia extracts on IOP in a rabbit model: (**a**) the ΔIOP (IOPtime-IOP0) of time course on the normotensive rabbit eye model. Dark blue is vehicle, light blue is control, red is 2.5 mg/kg CCME, green is 25 mg/kg CCME, and purple is Timolol. * *p* < 0.05, ** *p* < 0.01 and *** *p* < 0.001 versus vehicle control group; (**b**) the percentage of ΔIOP on the ocular normotensive rabbit model. Dark blue is vehicle, light blue is control, red is 2.5 mg/kg CCME, green is 25 mg/kg CCME, and purple is Timolol.

**Figure 5 molecules-27-00707-f005:**
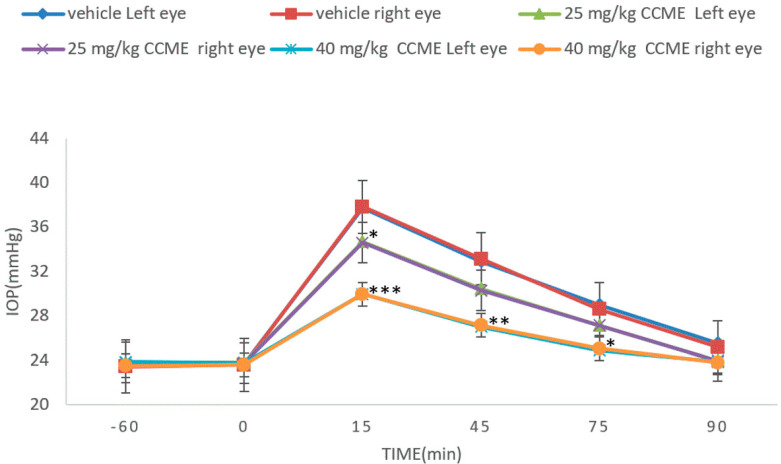
Anti-hypertensive effect of *C. cicadae* mycelia extract (CCME) in dextrose-induced acute glaucoma rabbit (A45—25 mg/kg/b.w.; B—40 mg/kg/b.w.) *n* = 4, mean ± SEM. * *p* < 0.05, ** *p* < 0.01 and *** *p* < 0.001, compared with solvent control group.

**Figure 6 molecules-27-00707-f006:**
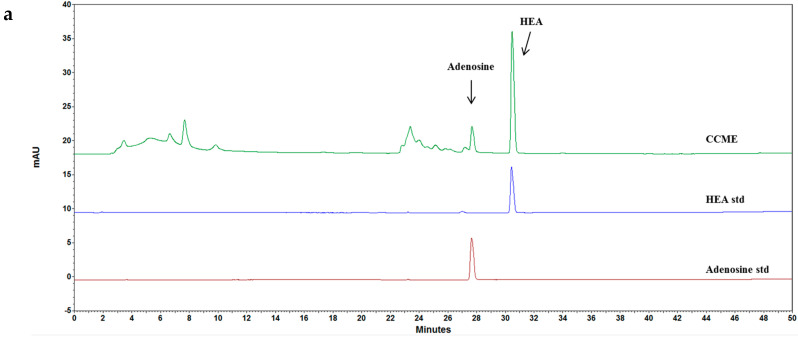
(**a**) HPLC of adenosine and HEA extracted from C. cicadae with retention times labeled at 27.7 min for adenosine and 30.8 min for HEA. Twenty microliters of adenosine or HEA; (**b**) LC-QTOF/MS spectra of adenosine with parental ion detected at *m*/*z* 268.10440; HEA with parental ion detected at *m*/*z* 310.11758.

**Figure 7 molecules-27-00707-f007:**
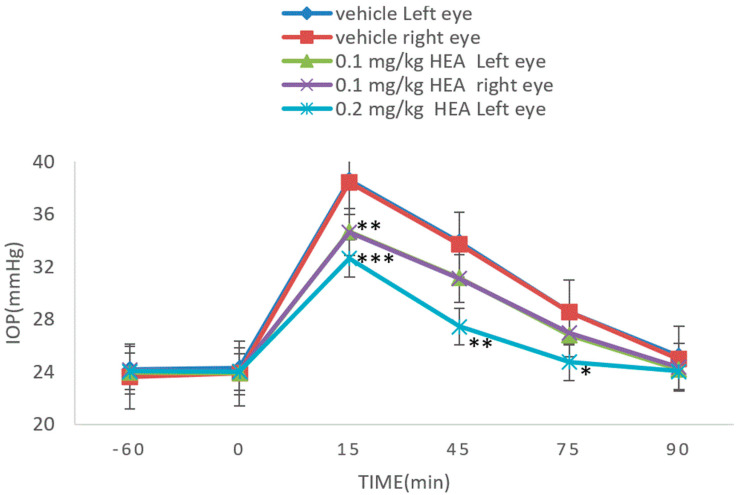
Anti-hypertensive IOP effect of HEA in dextrose induced acute glaucoma rabbit (A = 0.1 mg/kg/b.w., B = 0.2 mg/kg/b.w., *n* = 4). * *p* < 0.05, ** *p* < 0.01 and *** *p* < 0.001, compared with solvent control group.

**Table 1 molecules-27-00707-t001:** ROCK kinase inhibition assay for CCME.

Conc	Inhibition of %
CCME 0.25 μg/mL	4 ± 4
CCME 2.5 μg/mL	29 ± 4
CCME 25 μg/mL	4 ± 2
K-115 21.4 ± 1.6 nM	50

## Data Availability

All data can be assessed from L.Y. Lee via the email address.

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
