# Peer review of "Lowering the Intraocular Pressure in Rats and Rabbits by Cordyceps cicadae Extract and Its Active Compounds"

_molecules, 2022, doi:10.3390/molecules27030707_

Round 1
Reviewer 1 Report
- The main subtitles of the manuscript are anachronic. We suggest the authors to respect the proper order “Introduction, Materials and Methods, Results, Discussion and Conclusions”.
- From page 2, row 58, the authors didn’t use “justify” as page formatting.
- In page 1, row 15, the expression “to have a therapeut Glaucoma” contains phrasing errors, and the main idea is non-understandable. Please rephrase the sentence.
- In page 1, row 33, the phrase “C. cicadae is an entomopathogenic fungus that produces a fruit on the head of the host (Cicada flammata)” doesn’t describe exactly the effect of the fungus. Please rephrase and add more details.
- In page 1, row 43, the citation “[10,11,12,13,14,15,16,17,18,19]” should be noted as “[10-19].
- In reference 1, page 13, please remove the word “reference” added in front of the citation.
- Reference 28 is missing from the text.
- Some references do not meet the requirements of the journal. For example, the name of the journal should be abbreviated, in bibliography 13 it appears in full (Journal of Food and Nutrition Research instead of J. Food Nutr. Res.). Please check all references to be in accordance with the requirements.
- Please improve the conclusions of the manuscript.
The manuscript “Lowering the intraocular pressure in rats and rabbits by 2 Cordyceps cicadae extract and its active compounds” requires minor revision in terms of English spelling and coherence.
Author Response
- The main subtitles of the manuscript are anachronic. We suggest the authors to respect the proper order “Introduction, Materials and Methods, Results, Discussion and Conclusions”.
Reply 1: I should have checked the format of article more thoroughly. The format of the article has been modified.
- From page 2, row 58, the authors didn’t use “justify” as page formatting.
Reply 2: The format of the article has been modified.
- In page 1, row 15, the expression “to have a therapeut Glaucoma” contains phrasing errors, and the main idea is non-understandable. Please rephrase the sentence.
Reply 3: Modification completed.
- In page 1, row 33, the phrase “C. cicadae is an entomopathogenic fungus that produces a fruit on the head of the host (Cicada flammata)” doesn’t describe exactly the effect of the fungus. Please rephrase and add more details.
Reply 4: Modification completed.
- In page 1, row 43, the citation “[10,11,12,13,14,15,16,17,18,19]” should be noted as “[10-19].
Reply 5: Modification completed.
- In reference 1, page 13, please remove the word “reference” added in front of the citation.
Reply 6: Modification completed.
- Reference 28 is missing from the text.
Reply 7: Modification completed.
- Some references do not meet the requirements of the journal. For example, the name of the journal should be abbreviated, in bibliography 13 it appears in full (Journal of Food and Nutrition Research instead of J. Food Nutr. Res.). Please check all references to be in accordance with the requirements.
Reply 8: Modification completed.
- Please improve the conclusions of the manuscript.
Reply 9: Modification completed.

Reviewer 2 Report
The article describes the pharmacological potential of Cordyceps cicadae extract and its active compounds in minimizing intraocular pressure in rats and rabbits. The work demonstrated in this article is good work for readers. But the article has some flaws which should be removed before its final decision.
- The article overall is written in very poor English. It has taken a long for me to read one sentence 3-4 times then I was able to understand what the authors want to say. Even the first two lines of the abstract are very confusing. I don't know what the authors want to say. In the article during the center of sentences some alphabets are capital, some are small. It's very confusing. The Introduction is very confusing. The authors need to do a real major revision from start till end line by line very carefully.
- Fig.2, initial 2 microscopic pictures, the spelling of "Glaucoma" is written wrong on the picture. Kindly correct it. Furthermore, this fig. contains two subfigures which should be divided into "a", and "b" sections, and also mention this in the caption.
- The conclusion is not well written. As, the conclusion is one of the most important sections of the paper. So improve it. Further the last sentence of the conclusion. The authors said that "CCME is useful and can be used as medicine". Do the authors have proof for its safety, its toxicity, ADMET profile? how the authors can say that it can be used as medicine? IF the authors have proof of its toxicity, excretion, and everything then kindly provide it, otherwise remove this word.
- Kindly update your references. I just saw 2-3 references for 2020. Kindly up to date it with the latest references.
Author Response
- The article overall is written in very poor English. It has taken a long for me to read one sentence 3-4 times then I was able to understand what the authors want to say. Even the first two lines of the abstract are very confusing. I don't know what the authors want to say. In the article during the center of sentences some alphabets are capital, some are small. It's very confusing. The Introduction is very confusing. The authors need to do a real major revision from start till end line by line very carefully.
Reply 1: I should have checked the format of article more thoroughly. The format of the article has been modified.
2. Fig.2, initial 2 microscopic pictures, the spelling of "Glaucoma" is written wrong on the picture. Kindly correct it. Furthermore, this fig. contains two subfigures which should be divided into "a", and "b" sections, and also mention this in the caption.
Reply 2: I should have checked the format of article more thoroughly. The format of the article has been modified.
3. The conclusion is not well written. As, the conclusion is one of the most important sections of the paper. So improve it. Further the last sentence of the conclusion. The authors said that "CCME is useful and can be used as medicine". Do the authors have proof for its safety, its toxicity, ADMET profile? how the authors can say that it can be used as medicine? IF the authors have proof of its toxicity, excretion, and everything then kindly provide it, otherwise remove this word.
Reply 3: I should have checked the format of article more thoroughly. Conclusion content has been revised.
4. Kindly update your references. I just saw 2-3 references for 2020. Kindly up to date it with the latest references.
Reply 4: Thanks for reminding. We also provide references for 2021 .

Round 2
Reviewer 2 Report
The authors addressed all the questions very clearly and the paper is improved a lot as compared to before.